# Structure transfer and consolidation in visual implicit learning

Dominik Garber, József Fiser*

Department of Cognitive Science, Center for Cognitive Computation, Central European University, Vienna, Austria

## eLife Assessment

This study investigates the conditions under which abstract knowledge transfers to new learning. It presents **convincing** evidence across a number of behavioral experiments that when explicit awareness of learned statistical structure is present, knowledge can transfer immediately, but that otherwise similar transfer requires sleep-dependent consolidation. The **valuable** results provide new constraints on theories of transfer learning and consolidation.

**Abstract** Transfer learning, the re-application of previously learned higher-level regularities to novel input, is a key challenge in cognition. While previous empirical studies investigated human transfer learning in supervised or reinforcement learning for explicit knowledge, it is unknown whether such transfer occurs during naturally more common implicit and unsupervised learning and, if so, how it is related to memory consolidation. We compared the transfer of newly acquired explicit and implicit abstract knowledge during unsupervised learning by extending a visual statistical learning paradigm to a transfer learning context. We found transfer during unsupervised learning, but with important differences depending on the explicitness/implicitness of the acquired knowledge. Observers acquiring explicit knowledge during initial learning could transfer the learned structures immediately. In contrast, observers with the same amount but implicit knowledge showed the opposite effect, a structural interference during transfer. However, with sleep between the learning phases, implicit observers, while still remaining implicit, switched their behavior and showed the same pattern of transfer as explicit observers did. This effect was specific to sleep and not found after non-sleep consolidation. Our results highlight similarities and differences between explicit and implicit learning while acquiring generalizable higher-level knowledge and relying on consolidation for restructuring internal representations.

## Introduction

Recent years have seen a rise in interest in *transfer learning*, the application of previously learned regularities to novel input, as the effectiveness of this process is a key feature for biological and artificial intelligence alike (*Austerweil et al., 2019*; *Mark et al., 2020*; *Wang, 2021*). An important factor for successful transfer is the formation of abstract, generalizable knowledge transcending simple representation of the specific surface properties of the input (*Dekker et al., 2022*; *Samborska et al., 2022*; *Sun et al., 2023*; *Whittington et al., 2020*). A challenging aspect of this memory formation is balancing the search for new commonalities between different tasks or contexts and minimizing the interference between new experiences and old memories (*Flesch et al., 2018*; *Flesch et al., 2022*; *Flesch et al., 2023*). Previous empirical studies on transfer learning focused on the domains of supervised or reinforcement learning and mainly dealt with explicit knowledge. Although this allows for straightforward designs and ensures continuity with previous research, it ignores the fact that a large

*For correspondence:
fiserj@ceu.edu

Competing interest: The authors declare that no competing interests exist.

**eLife digest** People are constantly picking up patterns in the world around them, often without even trying. This is called implicit learning – a natural, unconscious process that helps us notice structure in things that might seem random at first. For example, we might start to recognize that certain shapes often appear together.

Scientists already know that when we learn something by actively thinking about it – called explicit learning – we can usually instantly apply that knowledge to new situations. That ability is known as generalization or transfer learning. However, it has been less clear whether generalization also happens through implicit learning and if time and sleep help shape or strengthen implicit memories over time.

To find out more, Garber and Fiser studied explicit and implicit visual learning tasks, in which 229 volunteers were presented with 20 abstract black shapes on a white background. During two learning phases, shapes were shown in fixed pairs, embedded within larger scenes composed of shapes– six pairs (same orientation) in phase one, and four pairs (two horizontal, two vertical) in phase two. Pair learning was assessed using a choice task. In each trial, participants viewed one real pair and one foil pair (mixed shapes from different pairs) and had to indicate the more familiar pair.

The experiments confirmed that when people learned with awareness, they could use the learned structure in new scenes right away. However, implicit learners did not transfer that knowledge immediately. Instead, they learned new patterns with different structures more easily than those that match the old ones. After 12 hours of sleep, though, they became able to apply the original structure. This suggests that sleep may play a specific role in reshaping unconscious knowledge to make it more flexible and useful.

Learning from patterns without thinking about it is important for everyone, from children and students to people starting new jobs or working in existing roles. A next step will be to test these findings in real-world settings using real-life situations and longer learning periods. Understanding how sleep timing and duration affect learning could help inform new strategies for improving education, training and even creative thinking.

part of ecologically relevant learning happens implicitly (*Reber, 1989*) and in the absence of supervision or reinforcement (*Hinton, 2014*; *Hinton, 2010*). It is, therefore, an open question whether and how humans' transfer learning operates during unsupervised learning and in the absence of explicit knowledge.

In the current study, we contrast implicit and explicit transfer learning in an unsupervised learning paradigm. Directly contrasting implicit and explicit learning is important as previous studies demonstrated that explicit and implicit forms of learning can show different outcomes and learning trajectories (*Ball et al., 2021*; *Bloch et al., 2016*; *Dale et al., 2012*; *Forano et al., 2021*; *Mathews et al., 1989*; *Poh and Taylor, 2019*). In addition, we investigated the effect of consolidation on implicit transfer learning at various timescales since the role of consolidation in structural abstraction and transfer has been demonstrated previously for explicit learning with supervision or feedback (*Chambers, 2017*; *Diekelmann and Born, 2010*; *Klinzing et al., 2019*; *Lerner and Gluck, 2019*; *Lewis and Durrant, 2011*; *Rasch and Born, 2013*) and since there exist interactions between the level of explicitness and consolidation (*Fischer et al., 2006*; *Liu et al., 2023*; *Robertson et al., 2004*; *Wagner et al., 2004*; *Zander et al., 2017*). Our paradigm builds on the classical spatial visual statistical learning (SVSL) design (*Fiser and Aslin, 2001*) and extends it to a transfer learning setup. In this new setup, participants proceed from learning reappearing patterns embedded in unsegmented input to abstracting their shared structure and finally applying this newly abstracted structure to novel unsegmented input. This paradigm, therefore, combines the challenge of segmentation based on co-occurrence statistics (statistical learning) with the abstraction and reapplication of shared structures (transfer learning), setting our study apart from previous studies on implicit abstraction or generalization (*Gómez, 2002*; *Marcus et al., 1999*; *Reber, 1967*).

Our results show that during unsupervised learning, the transfer of learned structures is immediately possible when explicit knowledge is acquired. In contrast, when only implicit knowledge is obtained, immediate transfer leads to an opposite effect, an interference between old knowledge

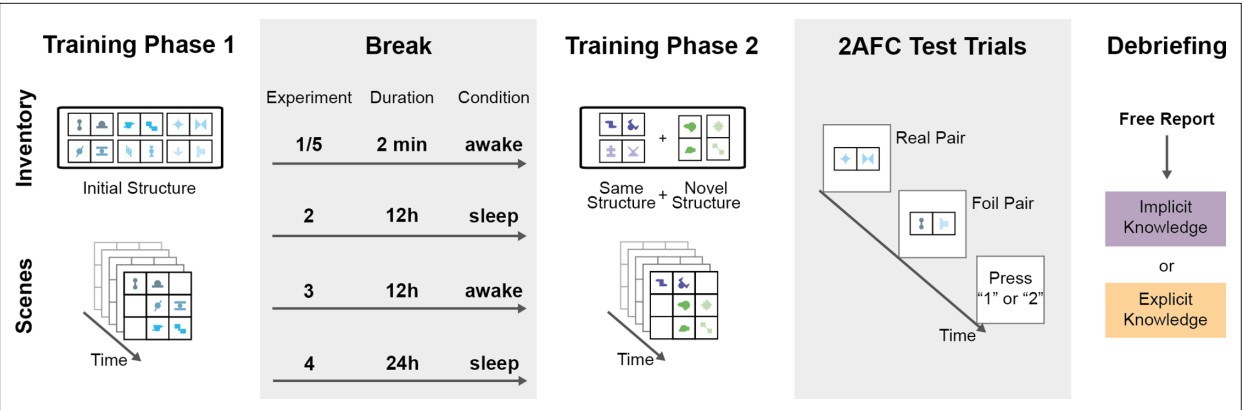

**Figure 1.** Overview of the experimental setup. **Training Phase 1:** Participants passively observed a stream of scenes made up of abstract shapes (*lower panel*). Unbeknown to the participants, shapes in the scene appeared only in pairs of fixed spatial configurations defined by the Inventory (*upper panel*). All pairs in Phase 1 had the same underlying structure of either horizontal or vertical orientation. The colors in the figure are only for illustration purposes; for the participants, all shapes were black. **Break:** After phase 1, there was a break that varied across the five experiments between 2 minutes and 24 hours. Participants spent the break either in asleep or awake condition. **Training Phase 2:** After the break, the participants were exposed to visual scenes made of a different set of abstract shapes. Half of the created pairs of the new inventory had horizontal, while the other half had vertical underlying structures. **2AFC Test Trials:** After Phase 2, participants completed a series of *2AFC Test Trials,* in which they had to decide if a real pair from the training phases or a foil pair, created by random combination of the shapes, was more familiar. **Debriefing:** Finally, participants answered open-ended questions about the experiment, which were used to assess whether they gained explicit knowledge about the presence of shape pairs.

and new learning. However, after 12-hour sleep consolidation following the learning of the first structure, while the knowledge still remains implicit, the same successful transfer of learned structures (i.e., generalization) occurs as in the explicit condition. Hence, our results extend the empirical findings on transfer learning to the domain of unsupervised implicit learning by showing that transfer learning is possible based on both explicit and implicit knowledge, albeit with different trajectories, and highlight two important causes for these different trajectories: explicitness and consolidation of the knowledge to be transferred.

## Results

### Visual statistical learning as a testbed for unsupervised hierarchical structure learning

Our experimental setup builds on the standard SVSL framework (*Fiser and Aslin, 2001*) and extends it to a transfer learning paradigm (*Figure 1*). In the traditional SVSL, participants passively watch a stream of scenes, each consisting of multiple shapes on a grid layout, which, unbeknown to the participant, are composed of a set of fixed shape pairs. The pairs are 'fixed' since the two constituent shapes of a pair always appear next to each other in the scenes in a fixed spatial relationship. Importantly, each pair abuts at least one other pair in the scenes, so the pairs are not separated visually from each other. Thus, the identities of the shape pairs need to be extracted by learning across multiple scenes to achieve successful segmentation. Adult and infant humans, as well as various other species, have been shown to automatically and implicitly extract the underlying pair structure of such scenes (*Fiser and Aslin, 2002*; *Lee et al., 2021*; *Santolin and Saffran, 2018*).

This paradigm is an ideal candidate for investigating unsupervised learning of higher-order structures as it not only allows learning the fixed pairs of shapes (chunks), but through these chunks, more abstract underlying structures can also be defined. One example of such a more abstract feature is the mean orientation of the learned chunks, where the orientation of a chunk is defined by whether the arrangement of the shapes within the pair is horizontal or vertical. Critically, these more abstract features are not observable properties of the scenes per se as there are no segmentation cues revealing the chunks instantaneously. Thus, only when the chunks are learned can their orientation emerge as a feature. Hence, the latent property of orientation can be extracted only in conjunction with learning the observable pair associations. We used this feature in our transfer learning version of the SVSL paradigm, but we went beyond simply assessing whether a previously presented latent

structure, such as the mean orientation of the learned chunks, became accessible to the observers. Instead, we measured how exposure to one type of abstract structure differentially influenced the acquisition of multiple types of structures later on.

Specifically, in the first training phase (Phase 1) in all of our experiments conducted online ('Materials and methods'), participants saw scenes composed from only horizontal pairs or only vertical pairs, depending on the assigned condition counterbalanced across participants. In a second training phase (Phase 2), following some delay, they saw scenes composed from both horizontal and vertical pairs and made from a novel set of shapes not used in Phase 1. In the test phase after Phase 2, participants completed a two-alternative forced-choice (2AFC) familiarity test indicating in each trial, whether they found more familiar a presented real pair (used in the training phases) or a foil pair (generated by randomly combining two shapes used in the training phase) (*Figure 1*). Due to a consistency effect detected in the responses for Phase 2 pairs, we used only half of the presented test trials of that phase for our analysis (see details in the 'Materials and methods' section). Our main question was how unsupervised learning of instances of a latent structure (e.g., horizontality by seeing only horizontal chunks) in Phase 1 influenced the unsupervised learning of instances of the same (horizontal) and a novel (vertical) structure in Phase 2. Our main measure of interest was the difference in learning the same vs. different types of pairs during Phase 2.

The explicitness of the acquired knowledge was assessed by using an exit survey at the very end of the experiment (see Appendix 1). Participants were labeled as 'explicit' if they gave any indication of being aware of reappearing fixed patterns/pairs in the scenes. It is worth noting that we excluded from our analysis the small number of participants who clearly reported awareness of the underlying structure (horizontality/verticality). Therefore, the participants we labeled 'explicit' in this study also had only far-from-perfect, partial knowledge about the structure of the scenes. In agreement with earlier visual statistical learning reports, the proportion of these excluded participants was 1–1.5% across experiments, and their exclusion did not change any of our results.

## Explicit learners generalize, while implicit learners show a structural novelty effect

Experiment 1 (n=226 after exclusions; see 'Materials and methods' for details) investigated the immediate transfer of knowledge about the higher-level structure between the two learning contexts by implementing a short 2-minute break between Phases 1 and 2 and, thereby, providing a baseline for the subsequent experiments. Both explicit and implicit learners in Experiment 1 performed above chance for pairs of the first training phase (Expl: M=67.9, SE=4.6, d=0.67, t(33)=3.93, p=0.002, BF = 70.8; Impl: M=55.0, SE=1.1, d=0.32, t(191)=4.46, p<0.001, BF=919). In addition, the explicit participants (n=34) performed significantly better than the implicit ones as shown by a Welch's *t*-test (n=192) (d=0.73, t(224)=2.75, p=0.009) (*Figure 2*). For Phase 2 pairs, explicit participants performed above chance for learning pairs that shared their higher-level orientation structure with that of pairs in Phase 1 (M=67.6, SE=6.3, d=0.48, t(33)=2.81, p=0.033, BF=5.0). The same participants showed some moderate learning of the new pairs with the novel (non-matching) structure as well, but this learning failed to reach significance (M=58.1, SE=5.6, d=0.25, t(33)=1.46, p=0.465, BF=0.48). In contrast to the explicit participants, implicit participants showed the opposite pattern, performing significantly above chance for pairs of the novel structure (M=57.8, SE=1.9, d=0.30, t(191)=4.16, p<0.001, BF=281) but demonstrated strong evidence that they did not learn pairs sharing the higher-level orientation structure with pairs in Phase 1 (M=49.1, SE=2.1, d=0.03, t(191)=–0.44, p=0.659, BF=0.09). The difference between the implicit participants' performances with same vs. different higher-order structures was statistically significant (M$_{diff}$=8.72, d=0.20, t(191)=2.78, p=0.030, BF=3.38). We note that, while the same difference was not significant for the explicit participants (d=0.19, t(33)=1.12, p=0.539, BF = 0.33), in their case, we observed a different kind of difference. There was a significant and medium-to-large size correlation between overall learning in the first training phase and learning of pairs with the same structure in the second training phase (r=0.45, p=0.008), while this correlation was absent between the same overall learning in the first learning phase and learning of pairs with novel structure in the second learning phase (r=−0.01, p=0.947). Thus, the qualitative pattern of the implicit participants' markedly better performance with novel structures was distinctively different from the pattern shown by the explicit participants, as indicated by the significant interaction between the two factors (participant type and structure type) of the two-factor mixed ANOVA results (F(1,224)=4.89, p=0.028,

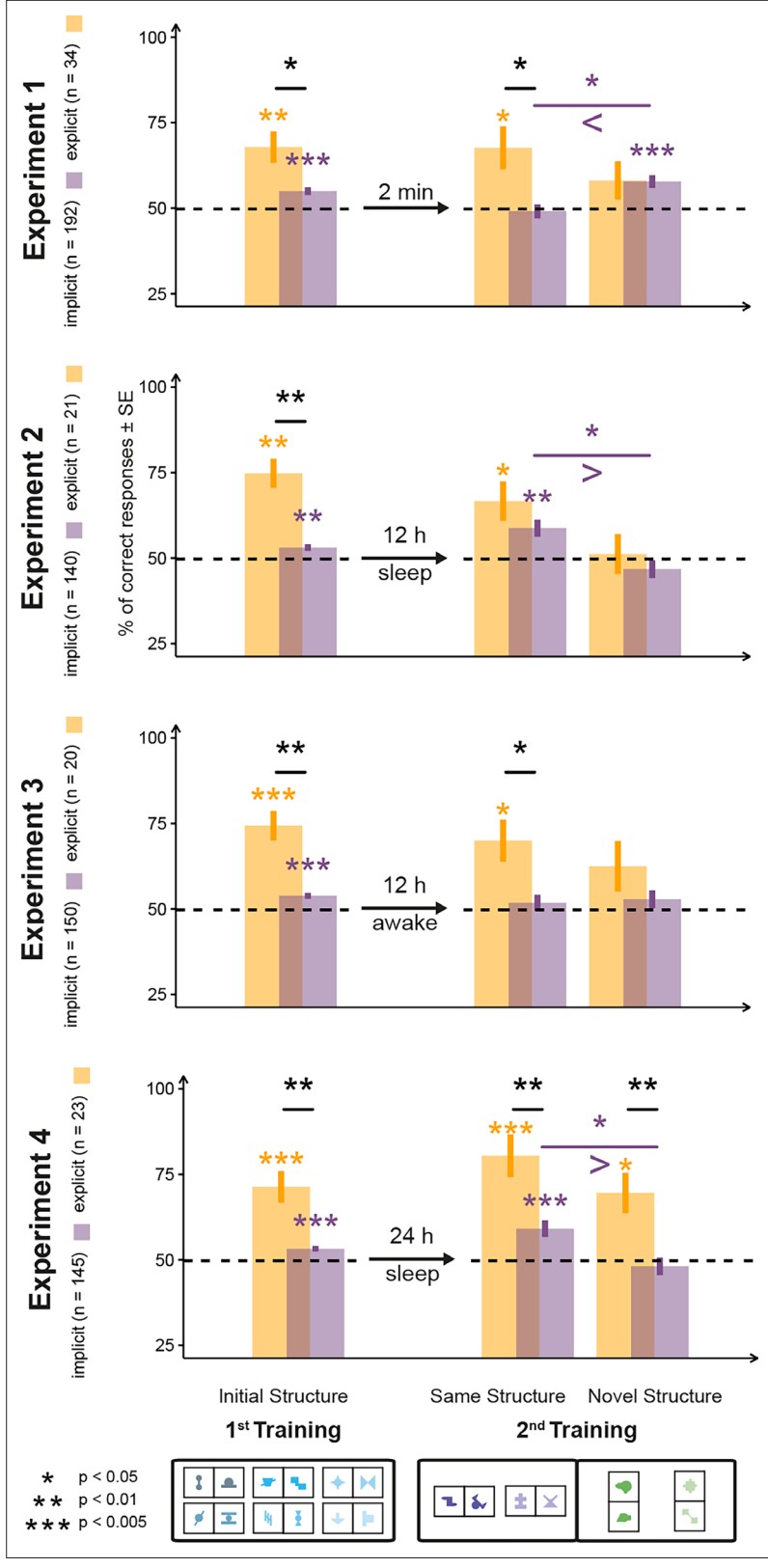

**Figure 2.** Results of familiarity tests in Experiments 1–4. Test results of the two-alternative forced-choice (2AFC) trials in all four experiments are grouped on the x-axis according to whether the trials used shapes of the first or the second training and, within the second training, whether the pair in the trial had the same or different orientation structure as inventory pairs in the first training. The y-axis represents the proportion of correct

*Figure 2 continued*

responses in the 2AFC test trials. Arrows and text between the test results related to the two trainings convey the condition and length of the break period. Bars represent SEM, color coding indicates implicit and explicit subgroups of the participants. The horizontal dotted line denotes chance performance. Asterisks above bars denote significance levels from chance, while above lines, significance level comparing two conditions below the tips of the line. Legend of significance levels is shown in the lower-left corner. Signs of inequality below the comparison in the second training indicate the direction of effect.

BF=31.6, $\eta_p^2$=0.02). These results imply that participants with more explicit acquired knowledge in Phase 1 effectively generalized the higher-level structure of this learned knowledge to novel situations and contexts, showing a 'structure transfer' effect. Meanwhile, implicit participants showed a 'structural novelty' effect that might be explained by a structure-level interference due to a larger representational overlap between previously learned and presently seen horizontal pairs than between previous horizontal and present vertical pairs.

To strengthen the reliability of our results, we conducted a conceptual replication of Experiment 1 using two types of diagonal pairs, which were again orthogonal to each other. The results again showed a structural novelty effect for implicit participants (see 'Supplementary Experiment 1').

## Consolidation enables implicit learners to generalize

To investigate the effect of consolidation on explicit and implicit learners' structural transfer in Experiment 2 (n=161 after exclusions; see 'Materials and methods' for details), we closely followed the design of Experiment 1 but introduced a 12-hour overnight consolidation phase between Phases 1 and 2. We found that while the performance of the explicit learners (n=21) after overnight sleep consolidation did not change drastically, it completely altered the pattern of behavior of implicit learners (n=140) as they demonstrated in this condition the same generalization as the explicit learners did in Experiment 1 (*Figure 2*).

Specifically, explicit learners (n=21) performed above chance for pairs of the same structure (M=66.7, SE=5.8, d=0.63, t(20)=2.87, p=0.038, BF=5.3) but not pairs of a novel structure (M=51.2, SE=5.8, d=0.04, t(20)=0.20, p=0.841, BF=0.23) in the test of Phase 2 pairs. Implicit learners (n=140) also performed above chance for pairs of the same structure (M=58.8, SE=2.5, d=0.30, t(139)=3.51, p=0.004, BF=37.6), but not for pairs of the novel structure (M=46.8, SE=2.6, d=0.11, t(139)=−1.24, p=0.435, BF=0.27). The performance difference between these two types of pairs was significant (M_diff=11.96, d=0.24, t(139)=2.82, p=0.027, BF=5.4). A direct comparison of implicit participants' performance in Experiments 1 and 2 showed that the participants performed higher after sleep for same structure pairs and lower for novel structure pairs (see next sections for detailed results). Overall, the results of Experiments 1 and 2 demonstrate that while explicit participants could immediately generalize the structure they learned, implicit participants required a consolidation period before being able to do the same.

## The effect of consolidation is specific to sleep

To clarify whether the effect observed in Experiment 2 is specific to sleep or just a general effect of consolidation, in Experiment 3 (n=170 after exclusions; see 'Materials and methods' for details), we used the same general procedure as in Experiment 2 but with the 12-hour consolidation phase occurring during the day. The results of the implicit learners (n=150) provided strong evidence against generalization as they performed at chance with both pairs of the same structure (M=51.8, SE=2.3, d=0.07, t(149)=0.80, p=0.999, BF=0.17) and pairs of a novel structure (M=52.8, SE=2.6, d=0.09, t(149)=1.08, p=0.999, BF=0.22) of Phase 2 (*Figure 2*). In contrast, participants with explicit knowledge (n=20) replicated the results of Experiments 1 and 2, with percent correct above chance for pairs of the same structure (M=70.0, SE=6.2, d=0.72, t(19)=3.24, p=0.026, BF = 10.4) but not for pairs of a novel structure (M=62.5, SE=7.4, d=0.38, t(19)=1.70, p=0.530, BF=0.78).

## The effect of sleep is not explained by a time-of-day effect

It has previously been suggested that an apparent effect of sleep shown in an AM-PM vs. PM-AM design can be based on the time of day at testing rather than sleep itself (*Tandoc et al., 2021*). To control for this potential confound, Experiment 4 (n=169 after exclusions; see 'Materials and methods'

for details) replicated Experiment 2 but with the second session delayed by 24 instead of 12 hours. Participants in this condition, therefore, had overnight sleep but were tested at the same time of day as the non-sleeping participants in Experiment 3. The results of Experiment 4 replicated the results of Experiment 2 for implicit learners, showing stronger learning for pairs of the same structure as compared to pairs of the novel structure ($M_{diff}$=11.04, d=0.23, t(144)=2.75, p=0.021, BF=4.4). We, therefore, conclude that the difference for implicit learners between our previous sleep and non-sleep conditions was not based on a time-of-day effect. For explicit learners, Experiment 4 showed no significant difference between learning for pairs of the same structure as compared to pairs of the novel structure ($M_{diff}$=10.78, d=0.26, t(22)=1.27, p=0.437, BF=0.44) and no significant correlation between learning pairs of the first learning phase and pairs of the novel structure (r=0.109, p=0.620). This suggests that the generalization effect found for explicit learners in the previous experiments is weakened during the longer consolidation phase of Experiment 4.

To compare directly the differential effect of type of consolidation on implicit structure learning, we entered the data of participants with implicit knowledge from Experiments 1–4 into a 4 × 2 ANOVA, with consolidation type (no consolidation, 12-hour-sleep, 12-hour-awake, and 24-hour-sleep consolidation) and pair type (same or novel structure) as factors. The obtained results showed the typical pattern of a cross-over interaction with no significant main effects (consolidation type: F(3,623)=0.18, p=0.910, BF=0.003, $\eta_p^2$=0.0009; pair type: F(1,623)=1.52, p=0.218, BF=0.17, $\eta_p^2$=0.002) but a significant interaction (F(3,623)=7.43, p<0.001, BF=1979, $\eta_p^2$=0.03). Post hoc analysis of achieved power in *gpower* (*Faul et al., 2007*) using the values for the interaction effect suggested that the achieved power was 0.99. Post hoc tests also revealed significant differences between the no-consolidation group (Experiment 1) and the two asleep-consolidation groups (Experiments 2 and 4), where the no-consolidation group showed stronger learning of novel structure pairs (Experiment 1 vs. Experiment 2: p=0.004, BF=44.3; Experiment 1 vs. Experiment 4: p=0.012; BF=12.7), while the asleep-consolidation groups showed stronger learning of same structure pairs (Experiment 1 vs. Experiment 2: p=0.015, BF=8.8; Experiment 1 vs. Experiment 4: p=0.011; BF = 14.3). No other significant differences were found. These results confirm for implicit learners a directly opposite pattern of generalization behavior between no-consolidation and asleep-consolidation conditions.

Additionally, to test for the presence of potential time-of-day effects in Experiment 1, we reanalyzed the data of Experiment 1 by taking into account the time point of testing. Both correlational and subgroup analyses found no indication of an effect of time of day on the pattern of structural transfer (see Appendix 1).

## The type of transfer depends on the quality of knowledge, not the quantity of knowledge

The different patterns of structural transfer for explicit and implicit participants could be based on either the quality of knowledge, that is, its explicitness, or the quantity of knowledge, that is, how much was learned during the first training phase. However, these two factors are confounded since the explicit participants typically performed higher for pairs of the first training phase. To address this confound, we conducted a matched sample analysis (*Ho et al., 2007*) in Experiments 1—4 to clarify which aspect of knowledge was responsible for our results. We selected a subsample of our implicit participants so that their accuracy performance matched that of the explicit participants for the first training phase and performed the same analyses on these subsampled populations as on the entire dataset in Experiments 1–4 (for details and a discussion of potential shortcomings, see Appendix 1). Matched implicit participants showed the same overall pattern of generalization behavior as the full sample of implicit participants for all four experiments (*Figure 3*), although this failed to reach significance for Experiment 4. Specifically, participants learned more pairs of the novel than of the same structure in Experiment 1 (p=0.012; BF=3.6), they learned more pairs of the same than of the novel structure in Experiment 2 (p<0.001, BF=127), and they showed no significant difference between learning the two types of pairs in Experiment 3 (p=0.214, BF=0.59) and Experiment 4 (p=0.304, BF=0.46). With the matched sample analysis, we drastically reduced the sample size of the implicit participants, therefore reducing the power to detect the small effects found for this group. However, we recovered the same descriptive pattern in the full data set and the matched sample groups in all experiments and failed to recover the same statistical significance only in the time-of-the-day control experiment. Therefore, we posit that these findings strongly support the notion that the difference in

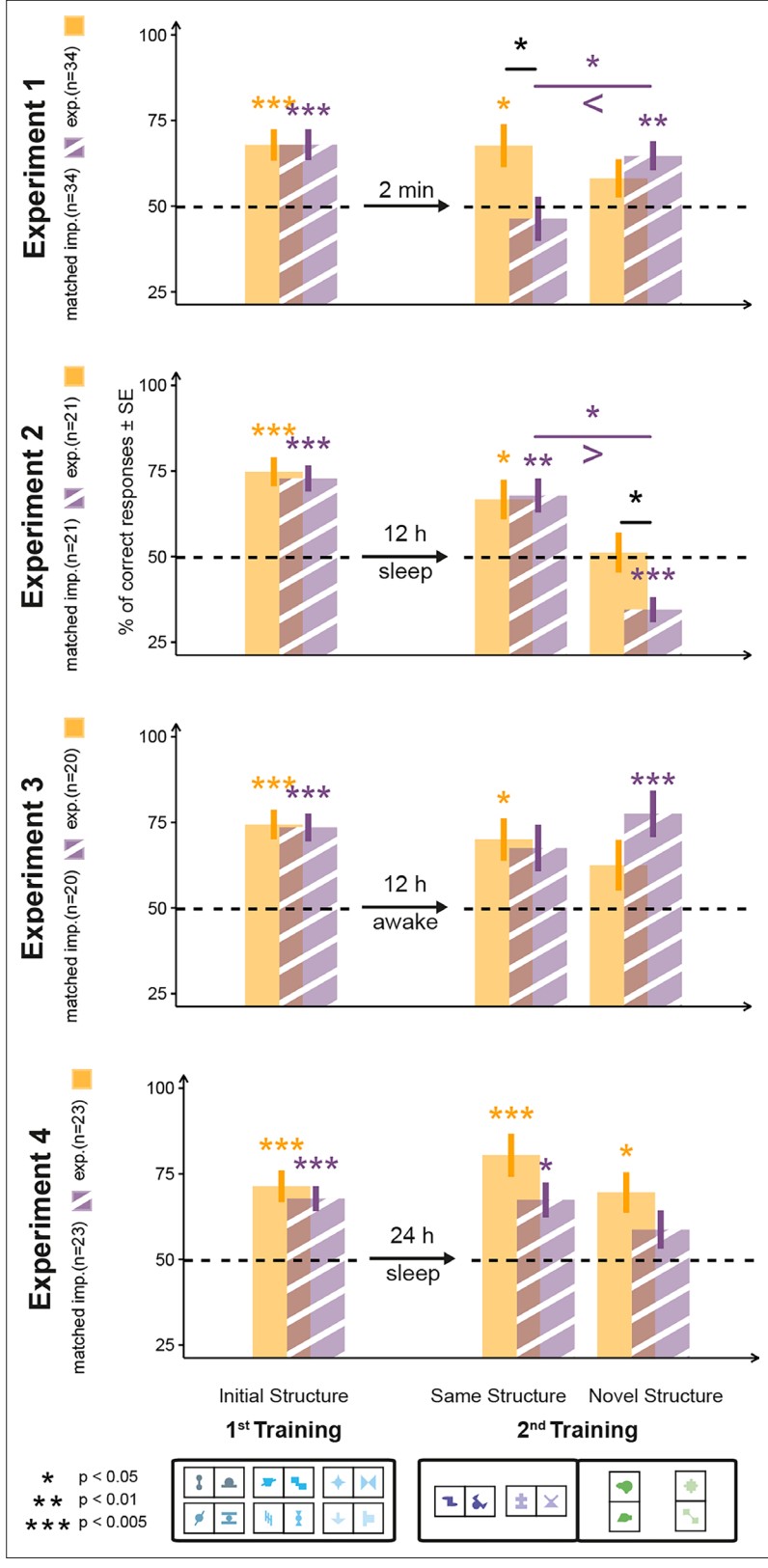

**Figure 3.** Matched sample analysis of Experiments 1–4. The structure of the figure is identical to that of *Figures 1 and 2*, with data for the explicit participants (orange bars) being the same as in *Figure 1*, while data for the implicit participants (striped purple bars) showing the subgroup of implicit participants from Experiment 1 whose combined performance matched the performance of the explicit participants on test trials of the first learning

*Figure 3 continued on next page*

*Figure 3 continued*

phase. The y-axes represent the proportion of correct responses in the 2AFC trials. Bars represent the mean (± SEM) for each type of pair (pairs of Phase 1 and same and novel structure pairs of Phase 2). The horizontal dashed line indicates chance performance.

the structural transfer is based predominantly on the quality of knowledge – its explicitness – not the exact level of learning.

## Explicitness and generalization can be induced by verbal instruction

The explicit–implicit parameterization in Experiment 1 was quasi-experimental rather than true experimental since the groups were formed naturally. Therefore, it is unclear whether it is indeed explicitness that enables generalization or whether the two groups of participants are different in other important ways (e.g., task engagement and attentional processes). To investigate this issue, in Experiment 5 (n=36 after exclusions), we rerun Experiment 1 with a new group of participants and with the single change in the protocol that explicitness of knowledge was induced via explicit verbal instructions stating that all shapes would be grouped into pairs. The results of Experiment 5 closely matched those of the explicit participants of Experiment 1 (*Appendix 1—figure 1*). The participants had an above-chance performance for Phase 1 (M=71.8, SE=3.6, d=1.01, t(35)=6.07, p<0.001, BF = 26,395) and for the same structure (M=72.9, SE=4.0, d=0.95, t(35)=5.69, p<0.001, BF=9058) but not novel structure pairs (M=52.8, SE=5.8, d=0.08, t(35)=0.48, p=0.634, BF=0.20) in Phase 2. The performance for pairs of the same and novel structure was significantly different (d=0.64, t(35)=3.83, p=0.001, BF=58.0). These results confirm that the generalization producing an immediate transfer between contexts can be easily induced by verbal instructions in any population, and therefore, the type of explicitness studied in our experiments is a likely candidate for the necessary condition for such a generalization.

In Experiments 1–4 and Supplementary Experiment 1, explicit participants showed the same descriptive pattern of higher performance for same over novel structure pairs, but this difference failed to reach significance within experiments. Therefore, we reanalyzed these results by collapsing the data over all these experiments. The results showed an overall higher performance on same structure pairs than novel structure pairs for explicit participants (d=0.34, t(233.99)=2.56, p=0.011, BF=3.12). Post hoc analysis of achieved power in *gpower* (*Faul et al., 2007*) effect suggested that the achieved power for this test was 0.96. Furthermore, over all experiments, explicit participants' performance for Phase 1 only significantly correlated with learning of same structure pairs (r=0.41, p<0.001, BF=4460), not novel structure pairs (r=0.14, p=0.137, BF=0.62). To validate the soundness of collapsing the data over experiments, we ran a 4 × 2 mixed ANOVA with factors Experiment and Test Type (Phase 1, Novel Structure, Same Structure). The results showed the expected significant main effect of Test Type (F(2,323)=5.52, p=0.004, BF=13.9, $\eta_p^2$=0.03) and a main effect of Experiment (F(4,323)=3.80, p=0.005, BF=0.677, $\eta_p^2$=0.04), but no significant interaction (F(8,323)=0.48, p=0.868, BF=0.02, $\eta_p^2$=0.01) supporting the notion that the data across experiments showed the same qualitative pattern.

## Discussion

Our findings demonstrated for the first time an interaction of consolidation and explicitness of knowledge in humans' unsupervised transfer learning. Participants with explicit knowledge are able to immediately transfer structural knowledge from one learning context to another. In contrast, participants with implicit knowledge show a structural novelty effect in immediate transfer. Only after a phase of sleep consolidation are they able to generalize from one learning context to another.

Our interpretation of these results is that sleep consolidation leads to an internal redescription of the acquired knowledge by factorization, that is, by representing the feature of orientation as a useful summary statistic in its own right. This redescription of the input with a hierarchical model using a higher-level representation of the underlying orientation structure emerges as a complement to the chunk representation learned by statistical learning during exposure. Without such a redescription, the previously learned patterns in Phase 1 interfere with new ones in Phase 2. When redescription occurs, the abstract knowledge at the higher level of the hierarchy can be generalized, that is, it can be used during the processing and segmentation of new input. Instead of a simple feed-forward

hierarchy progressing from specific to abstract, our interpretation implies a hierarchical system with bi-directional interactions: learning some representations at a higher level of abstraction (orientation) constrains the learning of new representations of lower-level features (actual chunks) later in time. Thus, our approach to hierarchy is different from earlier treatments of SL in hierarchical systems that focused on hierarchies of composition (*Lee et al., 2021*) or hierarchies that arbitrated between competing learning systems (*Maheu et al., 2022*).

One notable consequence of the emergence of such hierarchies is linking SL more to the emergence of object representations. Previous studies demonstrated that statistical learning is not based solely on momentarily observable co-occurrence statistics but is also biased by prior knowledge. However, these studies demonstrate various effects of prior knowledge acquired over a lifetime (*Stärk et al., 2023*) or shorter period (*Antovich and Graf Estes, 2023*; *Chough and Zinszer, 2022*; *Kóbor et al., 2020*), provided explicitly (*Arciuli et al., 2014*), or facilitated by segmentation cues (*Mueller et al., 2020*). They also focused mostly on the language domain or sequential streams and either relied on an uncontrolled amount of long-term knowledge or investigated token-level effects. In contrast, the current study investigated structure-related effects that could not be explained by the transfer of token-level knowledge, and it strictly controlled the amount of previous exposure to the target structure. Our results demonstrate that similar to objects (*Ashby and Maddox, 2005*; *Richler and Palmeri, 2014*), chunks acquired during SL are not represented simply as an unrelated set of inventory elements but organized into higher-level categories based on the common underlying structures – for example, the category of horizontal pairs. Thus, our results build a bridge between classical SL and category learning, and, in line with earlier findings (*Lengyel et al., 2019*; *Lengyel et al., 2021*), it supports the idea that SL is a vehicle for acquiring object-like representations.

Previous studies on statistical learning and consolidation focused on how much time or sleep helps with the stabilization or improvement of specific memory traces and obtained mixed results, with some studies showing clear effects (*Durrant et al., 2013*; *Durrant et al., 2016*; *Hennies et al., 2014*; *Lewis and Durrant, 2011*) and others showing no or very limited effects of sleep (*Arciuli and Simpson, 2012*; *Hallgató et al., 2013*; *Kim et al., 2009*; *McDevitt et al., 2022*; *Nemeth et al., 2010*; *Quentin et al., 2021*; *Simor et al., 2018*). Our results are in line with the latter set of studies, showing no evidence of a direct effect of sleep on statistical learning per se; that is, no effect on the learning of specific item-item associations. In addition, most of these studies did not consider the actual state of the acquired knowledge (but see *Liu et al., 2023*). In contrast, our study specifically focused on the difference between explicit and implicit learners to disentangle the complex nature of consolidation, and we used the ability to generalize structure between learning contexts as our primary measure. This approach allowed us to compare our findings to those reported in the domain of explicit learning (*Diekelmann and Born, 2010*; *Lerner and Gluck, 2019*; *Lewis and Durrant, 2011*; *Stickgold and Walker, 2013*), under names such as schemas (*Lewis and Durrant, 2011*), rules (*Lerner and Gluck, 2019*), insight (*Wagner et al., 2004*), and gist (*Lutz et al., 2017*), and uncover a surprising similarity between consolidation benefits based on abstraction and generalization in the two domains. In particular, similarly to observers who needed sleep to 'discover' abstract learning rules in an explicit task (*Stickgold and Walker, 2013*), our implicit learners required sleep to 'uncover' abstract higher-level descriptors of the observed structures and utilize them in subsequent implicit learning. This parallel in behavior suggests an analogous process for integrating and using implicitly and explicitly acquired new information at different levels of the existing internal representation.

A potential shortcoming of the current study is that, for practical reasons and due to the online nature of the experiments, the assignment of participants to the different consolidation conditions was not fully randomized. Instead, the various studies were made available at different points in time (i.e., data collection for Experiment 2 commenced after data collection for Experiment 1 was finished, etc.). This could, in principle, lead to a sampling bias due to preferences for conditions. Although we do not believe that it is likely that such a sampling bias on its own could produce the observed results in lieu of full experimental control, we cannot rule it out entirely.

A further uncontrolled factor in the current study is the exact timing of participants' acquisition of explicit knowledge. As we only collected the relevant information after the final test phase, we do not know when participants gained explicit knowledge during the experiment. However, assessing explicitness at the end of the study was a deliberate decision as the shortcoming of this method appears less severe than the potential of accidentally inducing explicitness by conducting the exit

survey earlier. This danger of inducing explicitness has been amply demonstrated by the results of Experiment 5. Nevertheless, future studies could use different methods of checking for explicitness at multiple points during the experimental procedure to gain a more refined view on the emergence of explicitness and its effects on behavior during subsequent learning. As a final point, previous studies showed that different measures of Statistical Learning can produce diverging results while assessing participants' levels of learning (*Bays et al., 2016*; *Kiai and Melloni, 2021*; *Liu et al., 2023*). In principle, it is possible that replicating the experiments of the current study with alternative measures of SL would yield somewhat different results, both quantitatively and qualitatively.

Our results in Experiment 1 suggest that for implicit participants learning in Phase 2 was subject to structure-specific proactive interference in the absence of consolidation as learning of same-structure pairs was hindered by the previous learning of pairs in Phase 1. Note that these findings do not exclude the possibility of simultaneously active retroactive interference. To establish if such an effect was present in the current study would require a further control condition of participants receiving only the Phase 1 training. We did not investigate this condition since the effect of retroactive interference was not the focus of the current study, and any obtained results would not alter our conclusions.

Previous statistical learning experiments reported that quantitative performance improves by explicit instructions (*Arciuli et al., 2014*) and that performances in the presence and absence of explicit instructions converge over longer exposure (*Arató et al., 2020*). Our results in Experiment 5 are in line with those findings by showing a stronger performance after explicit instructions about the task structure and a convergence between the outcomes of explicit and implicit learning. However, we also go beyond these findings by demonstrating the vanishing of differences after sleep consolidation even in cases when the difference between the initial implicit and explicit learning results prior to consolidation is qualitative rather than just numerically different.

In summary, our work builds multiple bridges between different domains of cognition. First, it relates low-level correlational learning and higher-level structure learning in an unsupervised transfer learning setup that is separate from the classical problem- or task-solving paradigms (*Ellenbogen et al., 2007*; *Fischer et al., 2006*; *Wagner et al., 2004*) yet can be readily interpreted in those frameworks. This extends the definition of higher-level structure to a domain where explicit tasks and thoughts are not defined. Second, it reports a new repulsion–attraction effect at the level of implicit structure transfer, namely a tradeoff between generalization and interference depending on the presence or absence of sleep-consolidation processes. Previously, such repulsion–attraction effects have been found either within immediate, implicit, and low-level perceptual processes but without linking the effect to sleep (*Daelli et al., 2010*; *Fritsche et al., 2020*; *Wei and Stocker, 2015*) or within high-level categorization setups that did investigate sleep effects but in the context of explicit tasks (*Diekelmann and Born, 2010*; *Djonlagic et al., 2009*; *Lerner and Gluck, 2019*; *Lewis and Durrant, 2011*; *Stickgold and Walker, 2013*). The converging sleep-related effects across implicit and explicit contexts might indicate the existence of a more general factorization mechanism in the brain that works continuously and at multiple levels during perceptual and cognitive processes following the same computational principles. Confirming this conjecture will require further investigation into the interplay between learning and consolidation in humans. Our approach can contribute to resolving this issue by facilitating an integrated exploration of how the brain learns complex internal representations implicitly and uses them in a recursive manner.

## Materials and methods
### Experiment 1
#### Participants

In total, 251 participants (92 females, age: mean=28.0, mode=25, SD=9.5) were recruited via prolific. co. All participants had normal or corrected-to-normal vision. The study was approved by the Hungarian United Ethical Review Committee for Research in Psychology (EPKEB), and all participants provided informed consent. Additional explicit consent for data sharing was not necessary as the data is fully anonymized for all reported experiments. The hourly compensation was £6.3. The sample size was chosen to achieve 80% power for expected small effect sizes (d=0.2) in paired *t*-tests (needed sample for the implicit subgroup according to power analysis = 198.15) and to account for exclusions.

The study protocol of Experiment 1 and all subsequent experiments was not preregistered before data collection and analysis.

Based on pilot data, we choose 20 seconds combined response time for both attention checks as the cut-off value for inclusion. Nineteen participants were rejected for failing this criterion. Response bias was defined as the proportion with which participants used one of the two response options ('1' and '2'), and participants who were 2.5 SD away from the mean were excluded. Three participants were excluded for failing this criterion. This left us with 229 participants after exclusions. Based on the open responses at the end of the experiment, participants were categorized into one of three groups. Participants who reported no knowledge of pairs were counted as implicit (n=192), participants who reported knowledge of the presence of pairs were counted as explicit (n=34), and participants who also reported the underlying horizontal/vertical structure were excluded from analysis as they were too few for meaningful analysis (n=3). See Appendix 1 for details on the exit survey and example responses.

## Materials

The stimuli were taken from *Fiser and Aslin, 2001* and consisted of 20 abstract black shapes on a white background (see *Figure 1*). The shapes were grouped to form six pairs of the same orientation (horizontal or vertical) for the first learning phase and four pairs, two horizontal and two vertical, for the second learning phase. The assignment of shapes to pairs was randomized for each participant. Scenes were created by placing three pairs together on a 3 × 3 grid without segmentation cues. In total, 160 scenes were created for the first and 48 for the second learning phase. In the second learning phase, each scene was used twice for a total of 96 presented scenes.

The study was conducted online via Prolific. As this was an online study, participants conducted it on their own computers using Google Chrome, Safari, or Opera browser. Only desktop and laptop computers were admissible, and no smartphones or tablets. Stimuli were presented using custom JavaScript code built on the jsPsych library (*de Leeuw, 2015*). As participants used different devices (screen size and resolution), the visual angle of the shapes was not the exact same for all participants. Instead, the 3 × 3 grid extended over 600 × 600 pixels and was centered in the middle of the screen. The remaining screen outside the grid was empty (white).

## Procedure

Participants passively observed 160 scenes in the first training phase. For half of the participants, these scenes contained only horizontal pairs (horizontal condition), and for the other half, only vertical pairs (vertical condition). Each scene was presented for 2 seconds with a 1-second interstimulus interval (ISI). After a 2-minute passive break, participants passively observed 96 scenes in the second training phase. Participants were not told about the presence of any structure in the scenes and were simply instructed to be attentive so that they could later answer simple questions. After half of each training phase, an attention check appears, asking participants to press the spacebar to continue. Response time for the attention check was recorded to detect inattentive participants. After the second training phase, participants had another two-minute passive break.

Following this, pair learning was tested with a 2AFC task. In each trial, participants saw a real shape pair from one of the training phases and a foil pair created by combining shapes from two different pairs of the same training phase. Real and foil pairs were presented after each other in the 3 × 3 grid for 2 seconds with a 1-second ISI. The order of real and foil pairs was randomized. Participants were asked to indicate which of the two was more familiar by pressing '1' or '2' on their keyboard. Participants first completed 16 trials using pairs from the second training phase, where each of the four real pairs of that training phase was tested against two different foil pairs twice. This was followed by 24 trials using pairs from the first training phase, where each of the six real pairs of that training phase was tested against four different foil pairs. Overall, all real and foil pairs were used the same number of times during the test phase. Finally, participants answered five open questions about their beliefs about the experiment and their knowledge of pair structure (see Appendix 1 for details).

## Additional analyses

Test trials for the second training phase were scored separately for pairs of the same structure as in the first training phase and pairs of the novel structure. Explorative analysis of the trials of the second

training phase revealed a strong negative correlation between trials of the novel and the same structure ($r=-0.432$, p<0.001). As the foil pairs used for one type of structure were created by recombining shapes of pairs of the other structure, this could be a type of consistency effect where participants tend to choose the same shapes, independent of pair knowledge. Specifically, we assumed that in addition to applying the weak pair knowledge acquired during the training phase, participants also considered the specific shapes seen in a test trial for their response. In other words, after a participant decided against a specific foil pair in an early test trial, they were less likely to pick an option using these same shapes in later test trials. Due to the foil pairs always being constructed of only shapes of pairs of the opposite structure, this would lead to participants developing a bias for only one type of structure (i.e., one set of shapes) in the course of the testing phase. If this is the case, we expect to see an increase in this negative correlation as participants complete more test trials. Indeed, for the first eight trials, this correlation is $r=-0.209$, while it increases to $r=-0.435$ in the second eight trials. Based on this exploration, we included only the first eight trials in all the following analyses to minimize this consistency effect. To keep comparability, the following experiments use the same number of test trials, but we again only analyze the first eight.

The data was collapsed over vertical and horizontal conditions for all further analysis, as a 3 × 2 mixed ANOVA with *test type* (levels: training 1, same structure, novel structure) as within-subject factor and *condition* (levels: horizontal, vertical) as between-subject factor showed no significant main effect of condition (F(1, 224)=0.776, p=0.379, BF=0.08, $\eta_p^2=0.003$) and no significant *test type–condition* interaction (F(2,448)=0.199, p=0.820, BF=0.04, $\eta_p^2=0.001$).

Bayes factors (BFs) reported for the ANOVA here and throughout the text are based on Bayesian ANOVAs using the *BayesFactor* R package, realizing Bayesian tests with models analogous to the frequentist counterpart, and employing a JZS prior (*Rouder et al., 2012*). BFs reported for *t*-tests throughout the text were calculated with the same package, following the same logic, and again employing the JZS prior unless specified otherwise. All other settings for the R functions of the *BayesFactor* package were left at the default values, unless specified otherwise. For the few Welch's *t*-test applied in the study, no BFs are reported as there is no readily available implementation for BF calculation for Welch's *t*-test. p-Values reported throughout the text were subject to experiment-wise correction for multiple comparisons using the Holm–Bonferroni method. All conducted tests are two-sided, unless specified otherwise.

We found a significant, medium to large correlation between learning in the first training phase and learning pairs of the same structure in the second training phase ($r=0.45$, p=0.008) for the explicit participants. Such a correlation was not observed between learning in the first learning phase and learning pairs of the novel structure in the second learning phase ($r=-0.01$, p=0.947). For implicit participants, neither of the two correlations was significant (*same structure*: $r=0.01$, p=0.853; *novel structure*: $r=0.04$, p=0.597); likely because the small effect sizes away from chance for this group lead to high noise in the data.

The different patterns of transfer behavior for explicit and implicit participants were also confirmed when entering the data into a 2 × 2 mixed ANOVA with factors *participant type* (explicit or implicit) and *structure type* (novel or same). The results show a significant main effect of *participant type* (F(1, 224)=8.03, p=0.005, BF=2.1, $\eta_p^2=0.03$) and of *structure type* (F(1, 224)=4.05, p=0.045, BF=1.2, $\eta_p^2=0.02$) as well as a significant interaction of both (F(1,224)=4.89, p=0.028, BF=31.6, $\eta_p^2=0.02$). Post hoc analysis of achieved power in *gpower* (*Faul et al., 2007*) using the values for the interaction effect and the correlation among the repeated measures reported above suggested that the achieved power for this test was 0.79.

To test for a possible time-of-day effect in learning or generalization, we correlated test performance with the hour of the day at which participants completed the experiment. There were no significant correlations for pairs of the same structure (explicit participants: $r=-0.03$, p=0.882; implicit participants: $r=0.01$, p=0.893) or pairs of the novel structure (explicit participants: $r=0.13$, p=0.477; implicit participants: $r=-0.05$, p=0.458). Additionally, we looked separately at groups of participants completing the experiment early in the day (7–11 am) and late in the day (7–11 pm). For implicit participants, there was no significant difference between participants that participated early (n=23) or late (n=22) as a 2 × 2 mixed ANOVA with *hour-of-day* and *test type* as factors showed no significant main effect of *hour-of-day* (F(1,43)=0.019, p=0.892, BF=0.26) and no significant *hour-of-day–test type* interaction (F(1,43)=0.095, p=0.759, BF=0.31).

## Experiment 2
### Participants
In total, 259 participants (127 females, age: mean=25.6, mode=21, SD=8.5) were recruited via prolific. co. All participants had normal or corrected-to-normal vision. The sample size was chosen to match that of Experiment 1. The study was approved by the Psychological Research Ethics Board of the Central European University (Approval Identifier: PREBO_2021/06_01), and all participants provided informed consent. The hourly compensation was £6.3. To ensure that participants have overnight sleep during the experiment as intended, several constraints and checks were implemented (see 'Sleep during consolidation studies' section of Appendix 1).

### Materials
The materials of the main part of the experiment were identical to Experiment 1. Additionally, participants filled out the Pittsburgh Sleep Quality Index (PSQI) (*Buysse et al., 1989*) and the Groningen Sleep Quality Scale (GSQS) (*Meijman and Vries-Griever, 1988*).

### Procedure
The procedure within the main tasks was identical to Experiment 1. However, in this experiment, participants completed the first training phase in the evening at 9 pm, followed by the GSGS and PSQI questionnaires. 12 hours later in the morning at 9 am, they completed the second training phase, followed by all the test trials and finally, they completed the GSGS again.

### Additional analyses
BFs from Bayesian *t*-tests for implicit participants reported for Experiments 2, 3, 4, and Supplementary Experiment 1 used an *r*-scale parameter of 0.5 instead of the default $\sqrt{2}/2$. The *r*-scale parameter controls the prior over effect sizes and using 0.5 means increasing the prior probability of small effects. This decision was made to incorporate the shifted belief about expected effect sizes for implicit participants after observing the effects in Experiment 1.

As in Experiment 1, for explicit participants, the difference between pairs of the same and novel structure was not significant (d=0.39, t(20)=1.77, p=0.272, BF=0.86), but we did see a strong positive correlation between learning in the first learning phase and learning pairs of the same structure in the second learning phase (*r*=0.52, p=0.015). Such a correlation was again not observed between learning in the first learning phase and learning pairs of the novel structure in the second learning phase (*r*=0.27, p=0.232). For implicit participants, neither of the two correlations was significant (*same structure*: *r*=0.05, p=0.570; *novel structure*: *r*=−0.02, p=0.788); likely because the small effect sizes away from chance for this group lead to high noise in the data.

## Experiment 3
### Participants
In total, 275 participants (134 females, age: mean=28.9, mode=24, SD=9.8) were recruited via prolific. co. All participants had normal or corrected-to-normal vision. The sample size was chosen to match that of Experiment 1. The study was approved by the Psychological Research Ethics Board of the Central European University (Approval Identifier: PREBO_2021/06_01), and all participants provided informed consent. The hourly compensation was £6.3. To ensure that participants have overnight sleep during the experiment as intended, several constraints and checks were implemented (see 'Sleep during consolidation studies' section of Appendix 1).

### Materials
The materials were identical to those of Experiment 2.

### Procedure
The procedure was the same as Experiment 2, with the difference that the first session took place in the morning at 9 am and the second in the evening at 9 pm. As there was no night of sleep between the first and second sessions, participants filled out the GSQS only once in this experiment.

## Additional analyses

The results (*Figure 2*) showed above chance performance in the first training phase for participants with implicit knowledge (n=150) (M=53.9, SE=0.9, d=0.36, t(149)=4.46, p<0.001, BF=1020) and participants with explicit knowledge (n=20) (M=74.4, SE=4.3, d=1.27, t(19)=5.67, p<0.001, BF=1281). As in Experiment 1, for explicit participants, the difference between pairs of the same and novel structure was not significant (d=0.19, t(19)=0.86, p=0.999, BF=0.32), but we again saw a strong positive correlation between learning in the first learning phase and learning pairs of the same structure in the second learning phase (*r*=0.56, p=0.010). Such a correlation was not observed between learning in the first learning phase and learning pairs of the novel structure in the second learning phase (*r*=0.42, p=0.065). For implicit participants, the *same structure pairs* (*r*=0.07, p=0.430) correlation was not significant, while we saw a small significant correlation for the *novel structure* pairs (*r*=0.18, p=0.025).

## Experiment 4

### Participants

In total, 275 participants (129 females, age: mean=27.9, mode=23, SD=8.9) were recruited via prolific. co. All participants had normal or corrected-to-normal vision. The sample size was chosen to match that of Experiment 1. The study was approved by the Psychological Research Ethics Board of the Central European University (Approval Identifier: PREBO_2021/06_01), and all participants provided informed consent. The hourly compensation was £6.3. To ensure that participants have overnight sleep during the experiment as intended, several constraints and checks were implemented (see 'Sleep during consolidation studies' section of Appendix 1).

### Materials

The materials were identical to Experiment 2.

### Procedure

The procedure was identical to Experiment 2. However, in this experiment participants completed the first session in the evening at 9 pm, and the second session 24 hours later again in the evening at 9 pm.

### Additional analyses

The results (*Figure 2*) showed that participants with implicit knowledge (n=145) performed above chance for pairs of the first training phase (M=53.2, SE=0.8, d=0.34, t(144)=4.09, p=0.001, BF=260) and for pairs of the same structure (M=59.1, SE=2.4, d=0.31, t(144)=3.78, p=0.001, BF=89.4) but not pairs of a novel structure (M=48.1, SE=2.6, d=0.06, t(144)=−0.74, p=0.459, BF=0.17) in the second training phase. Participants with explicit knowledge (n=23) performed above chance for pairs of the first training phase (M=71.4, SE=4.6, d=0.9, t(22)=4.6, p=0.001, BF=217), as well as for pairs of the same structure (M=80.4, SE=6.3, d=1.01, t(22)=4.85, p=0.001, BF=347), and pairs of a novel structure (M=69.6, SE=5.9, d=0.69, t(22)=3.33, p=0.012, BF=13.6) in the second training phase. For explicit participants, we did not see a significant correlation between learning in the first learning phase and learning pairs of the same structure (*r*=0.109, p=0.620) or of the novel structure (*r*=0.111, p=0.613) in the second learning phase. For implicit participants, neither of the two correlations was significant as well (*same structure*: *r*=−0.01, p=0.894; *novel structure*: *r*=0.01, p=0.940).

## Experiment 5

### Participants

Forty participants (18 females, age: mean=28.4, mode=19, SD=11.8) were recruited via prolific.co. All participants had normal or corrected-to-normal vision. The sample size was chosen to approximately match the number of explicit participants in Experiment 1 after exclusions. The study was approved by the Psychological Research Ethics Board of the Central European University (Approval Identifier: PREBO_2021/06_01), and all participants provided informed consent. The hourly compensation was £6.3. To ensure that participants have overnight sleep during the experiment as intended, several constraints and checks were implemented (see 'Sleep during consolidation studies' section of Appendix 1).

## Materials
The materials were identical to Experiment 1.

## Procedure
The procedure was identical to Experiment 1, apart from the instructions. In this experiment, participants were told about the pair structure before the beginning of the experiment. Participants were told that whenever a specific shape appears in the grid a second specific shape will appear in a fixed position near it.

# Additional information

### Funding

| Funder | Grant reference number | Author |
|---|---|---|
| Austrian Science Fund | 10.55776/I6793 | József Fiser |
| National Science Foundation | PHY-2309135 | József Fiser |
| Gordon and Betty Moore Foundation | 2919.02 | József Fiser |

The funders had no role in study design, data collection and interpretation, or the decision to submit the work for publication.

### Author contributions
Dominik Garber, Conceptualization, Data curation, Software, Formal analysis, Validation, Investigation, Visualization, Methodology, Writing – original draft, Writing – review and editing; József Fiser, Conceptualization, Supervision, Funding acquisition, Validation, Investigation, Visualization, Methodology, Writing – original draft, Project administration, Writing – review and editing

### Author ORCIDs
Dominik Garber ⬡ https://orcid.org/0000-0001-5869-6993
József Fiser ⬡ https://orcid.org/0000-0002-7064-0690

### Ethics
Human subjects: The study was approved by the Hungarian United Ethical Review Committee for Research in Psychology (EPKEB) and all participants provided informed consent. Consent of publishing then data was not collected because we did not publish any part of the data with identifiers of the participants.

Reviewer #1 (Public review): https://doi.org/10.7554/eLife.100785.4.sa1
Reviewer #2 (Public review): https://doi.org/10.7554/eLife.100785.4.sa2
Reviewer #3 (Public review): https://doi.org/10.7554/eLife.100785.4.sa3
Author response https://doi.org/10.7554/eLife.100785.4.sa4

# Additional files

### Supplementary files
MDAR checklist

### Data availability
The experimental data that support the findings of this study are available on OSF: https://osf.io/untbz.

The following dataset was generated:

| Author(s) | Year | Dataset title | Dataset URL | Database and Identifier |
|---|---|---|---|---|
| Garber D | 2025 | structure_sleep_implicit | https://osf.io/untbz | Open Science Framework, osf.io/untbz |

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

# Appendix 1

## Supplementary Experiment 1

### Participants

In total, 243 participants (128 females, age: mean=30.1, mode=24, SD=11.7) were recruited via prolific.co. The hourly compensation was £6.3. All participants had normal or corrected-to-normal vision. The sample size was chosen to match that of Experiment 1. The study was approved by the Psychological Research Ethics Board of the Central European University (Approval Identifier: PREBO_2021/06_01), and all participants provided informed consent.

### Materials

This experiment used the same materials as Experiment 1. However, the shapes were not grouped into horizontal and vertical pairs but into two orthogonal groups of oblique pairs. For one group of pairs, the second shape was always in the top right grid cell from the first shape, for the other group of pairs the second shape was always in the top-left grid cell. As such pairs allow for fewer unique combinations within a 3 × 3 grid, the scenes for this experiment were set in a 5 × 5 grid. The shapes mainly occupied the central 3 × 3 sub-grid, with one shape per scene being in the outer cells. All shapes appeared in the outer cells an equal number of times over all scenes. In order to ensure that the whole grid is visible on the participants' devices, the size was not set to a fixed pixel value. Instead, the size of one grid cell was set to 1/7 of the pixel height of the participant's screen. Therefore, the whole grid filled 5/7 of the screen height.

### Procedure

The procedure was identical to Experiment 1.

### Results

Overall, the data shows the same pattern as in Experiment 1 (see *Appendix 1—figure 1*, bottom panel). The results for the implicit participants (n=212) closely follow the results of Experiment 1. They perform above chance for pairs of the first training phase (M=53.9, SE=0.8, d=0.34, t(211)=4.99, p<0.001, BF=10,258) and for pairs of a novel structure (M=57.1, SE=1.7, d=0.28, t(211)=4.06, p<0.001, BF=245) but not pairs of the same structure (M=50.1, SE=1.8, d=0.03, t(211)=0.46, p=0.999, BF=0.12) in the second training phase. The performance for pairs of the same and novel structure is again significantly different (d=0.18, t(211)=2.67, p=0.049, BF=3.2). The results for explicit participants (n=12) show the same qualitative pattern as in Experiment 1, however, without reaching a significant difference from chance: first training phase (M=59.4, SE=4.8, d=0.56, t(11)=1.95, p=0.387, BF=1.2), novel structure (M=52.1, SE=10.4, d=0.06, t(11)=0.20, p=0.999, BF=0.29), and same structure (M=60.4, SE=7.8, d=0.39, t(11)=1.33, p=0.084, BF=0.59). This can be explained by the significantly smaller number of participants acquiring explicitness in Supplementary Experiment 1 compared to 1 ($X^2$=10.47, df=1, p=0.001, BF = 37.0), which results in diminished power for these tests.

## Sleep during consolidation studies

In order to assure that participants had overnight sleep during Experiments 2 and 4, and that they did not sleep during the day in Experiment 3, a number of constraints and checks were implemented. First, participants were not taken from the full global prolific pool but restricted to a number of European countries within the same time zones. Country of residence is one of the attributes of participants that prolific.co verifies. In order to roughly approximate the geographic distribution of participants in Experiment 1 (see *Appendix 1—figure 4*), we chose countries from two time zones. GMT±00:00 (countries: the UK and Portugal) and GMT+01:00 (Countries: Germany, France, Spain, Czech Republic, Denmark, Hungary, Italy, the Netherlands, Poland, Slovenia, Switzerland.) Second, as we do not expect Prolific's residence information to be perfectly predictive of where participants are while they conduct the experiment, participants were asked what the current time at their location is. Third, at the start of the second part of the experiment, participants were asked how much they slept between the first and second parts. This was used to exclude participants from Experiment 3 who slept during the day.

## Matched sample analysis

As reported in the main text, explicit participants show higher average learning in the first learning phase, which could be what enables the generalization of the learned structure. To test this idea, we conducted a matched sample analysis (*Ho et al., 2007*). The general idea of this analysis is to create a sub-sample of the implicit participants that perform like the explicit participants for the pre-training trials.

For all experiments, in a first step, we ran six applicable matching algorithms implemented in the MatchIt R package (*Ho et al., 2011*). The six so-created matched implicit samples were then compared to the original explicit sample according to four metrics: standardized mean difference, variance ratio, mean of the empirical cumulative density function, and maximum of the empirical cumulative density function. All values for all experiments can be seen in *Appendix 1—Tables 1–4*. 'Unbalanced' denotes the values for the full, non-matched implicit sample. All values for the used matching methods are evaluated as an improvement from those values. Standardized mean difference describes how far the mean of the matched sample is from the comparison sample (explicit participants); values closer to zero are better. The variance ratio is the ratio of the variances of the matched and the comparison sample; the best possible value is 1. The eCDF (empirical cumulative density function) contains more information than the mean and variance ratio as they capture the whole distribution of values. Two commonly used simple metrics based on the eCDF are the mean and maximum difference of the eCDFs of the matched and comparison group. Generally, values closer to zero are better. The best-fitting matching algorithm was not exactly aligned for all experiments. For consistency reasons, we chose the overall best-fitting method for all experiments: nearest neighbor matching with replacement. To provide a detailed overview of how this decision might have influenced our results, we plotted the results for all tested matching algorithms in *Appendix 1—figure 5*.

A potential shortcoming of the applied matching algorithm is that some participants will reach higher than 50% performance by chance. Therefore, some of the matched implicit participants may not actually have the same level of learning as the explicit participant they are matched to as they might have achieved their apparent level of performance by chance. However, there are two strong reasons why this is highly unlikely. First, it is obvious from comparing the results from our full sample analysis to the matched sample analysis that the qualitative pattern of results in Phase 2 is identical between the two groups, suggesting that Phase 2 performance is virtually independent of Phase 1 performance for implicit learners. This similarity in patterns weakens the assumption that the matched group behaves fundamentally differently from the full implicit group. Second, if participants in the matched implicit group performed better than average only by chance, it is difficult to argue that at the same time their performance across Same and Novel trials had a significantly non-random pattern that was the exact opposite of that shown by the Explicit participants.

## Experiment 1 results

We found that the matched sample showed a similar pattern for learning in the second training phase as the original full sample (see *Figure 3*). This is captured by a 2 × 2 ANOVA using the novel and same structure pairs for the original explicit and the matched implicit data showing a significant interaction ($F(1,87)=8.53$, $p=0.004$, BF=10.7, $\eta_p^2=0.09$) and post hoc comparisons show a significant difference between novel and same structure trials for the synthetic implicit data ($p=0.012$; BF=3.6). This analysis suggests that the difference between the two groups is not merely based on different strengths of learning in the first training phase.

## Experiment 2 results

As in Experiment 1, the matched sample showed the same type of pattern as the full sample (see *Figure 3*). As a critical analysis, we can see that for the matched implicit sample there is a significant difference between learning pairs of the same and of the novel structure ($d=0.98$, $t(20)=4.51$, $p<0.001$, BF=127), suggesting generalization of the structure.

## Experiment 3 results

As previously, the matched sample showed a similar type of pattern as the full sample (see *Figure 3*). Critically, we can see that for the matched implicit sample there is no significant difference between

learning pairs of the same and of the novel structure (d=0.29, t(19)=−1.29, p=0.214, BF=0.59), suggesting no generalization of the structure.

## Experiment 4 results

As in Experiment 1, the matched sample showed the same type of pattern as the full sample descriptively (see *Figure 3*). However, the critical analysis of the difference between learning pairs of the same and of the novel structure for the matched implicit sample failed to reach significance ($M_{diff}$=8.66, d=0.22, t(22)=1.05, p=0.304, BF=0.46).

**Appendix 1—table 1.** Experiment 1: overview of balance metrics for the used matching algorithms.

| Matching method | Standardized mean difference | Variance ratio | eCDF mean | eCDF max |
|---|---|---|---|---|
| Unbalanced | 0.486 | 2.92 | 0.169 | 0.358 |
| **NN with replacement** | **−0.002** | **0.95** | **0.003** | **0.059** |
| NN without replacement | 0.078 | 1.31 | 0.026 | 0.206 |
| Optimal pair matching | 0.078 | 1.31 | 0.026 | 0.206 |
| Optimal full matching | −0.018 | 1.05 | 0.012 | 0.059 |
| Coarsened exact matching | −0.021 | 1.12 | 0.021 | 0.118 |
| Subclassification | −0.049 | 1.19 | 0.020 | 0.059 |

eCDF, empirical cumulative density function.

**Appendix 1—table 2.** Experiment 2: overview of balance metrics for the used matching algorithms.

| Matching method | Standardized mean difference | Variance ratio | eCDF mean | eCDF max |
|---|---|---|---|---|
| Unbalanced | 1.125 | 2.909 | 0.273 | 0.548 |
| **NN with replacement** | **0.432** | **2.424** | **0.105** | **0.429** |
| NN without replacement | 0.103 | 1.089 | 0.030 | 0.238 |
| Optimal pair matching | 0.432 | 2.424 | 0.105 | 0.429 |
| Optimal full matching | 0.112 | 1.079 | 0.033 | 0.238 |
| Coarsened exact matching | 0.142 | 1.130 | 0.037 | 0.294 |
| Subclassification | 0.248 | 1.175 | 0.060 | 0.238 |

eCDF, empirical cumulative density function.

**Appendix 1—table 3.** Experiment 3: overview of balance metrics for the used matching algorithms.

| Matching method | Standardized mean difference | Variance ratio | eCDF mean | eCDF max |
|---|---|---|---|---|
| Unbalanced | 1.067 | 3.284 | 0.290 | 0.49 |
| **NN with replacement** | **0.293** | **2.260** | **0.079** | **0.30** |
| NN without replacement | 0.043 | 0.950 | 0.018 | 0.25 |
| Optimal pair matching | 0.293 | 2.260 | 0.079 | 0.30 |
| Optimal full matching | 0.068 | 1.073 | 0.023 | 0.25 |
| Coarsened exact matching | 0.043 | 1.161 | 0.023 | 0.25 |
| Subclassification | 0.089 | 1.106 | 0.039 | 0.25 |

eCDF, empirical cumulative density function.

**Appendix 1—table 4.** Experiment 4: overview of balance metrics for the used matching algorithms.

| Matching method | Standardized mean difference | Variance ratio | eCDF mean | eCDF max |
|---|---|---|---|---|
| Unbalanced | 0.819 | 5.352 | 0.222 | 0.494 |
| NN with replacement | 0.450 | 3.006 | 0.106 | 0.391 |
| NN without replacement | 0.164 | 1.327 | 0.043 | 0.304 |
| Optimal pair matching | 0.450 | 3.006 | 0.106 | 0.391 |
| Optimal full matching | 0.151 | 1.479 | 0.047 | 0.304 |
| Coarsened exact matching | 0.026 | 1.099 | 0.016 | 0.064 |
| Subclassification | 0.240 | 2.404 | 0.071 | 0.304 |

eCDF, empirical cumulative density function.

## Exit survey

The test trials of all experiments were followed by a set of open-ended questions. Two of these questions were jointly used to assess participants' explicitness of knowledge. Participants were labeled as 'explicit' if they gave any indication of being aware of reappearing fixed patterns/pairs in the scenes. Additionally, a small number of participants who clearly reported the underlying structure (horizontality/verticality) were excluded from the analysis.

### Questions:

1. "Please, explain with your own words what you think the experiment was about."
2. "In the first part of the experiment, where you only passively watched the screen: Did you notice any regularities in how the shapes were arranged? If yes, please describe them!"

### Example answers used for classification:

#### Reporting underlying structure:
- "The shapes had consistent pairs in the same row of the grid."
- "There were five pairs and all of them were matching vertically."

#### Explicit:
- Example of general pair knowledge: "I noticed the same two shapes were always together.'
- Example of specific pair knowledge: "The 'flag' was always paired with the 'sun', the 'square' with the 'arrow', I think."

#### Implicit:
- Example of no reported knowledge: "I tried to find some regularities, but I couldn't find any. Everything looked random to me."
- Example of no relevant knowledge: "I think that cross symbol was always in the top or bottom row, never in the middle. But I am probably wrong."

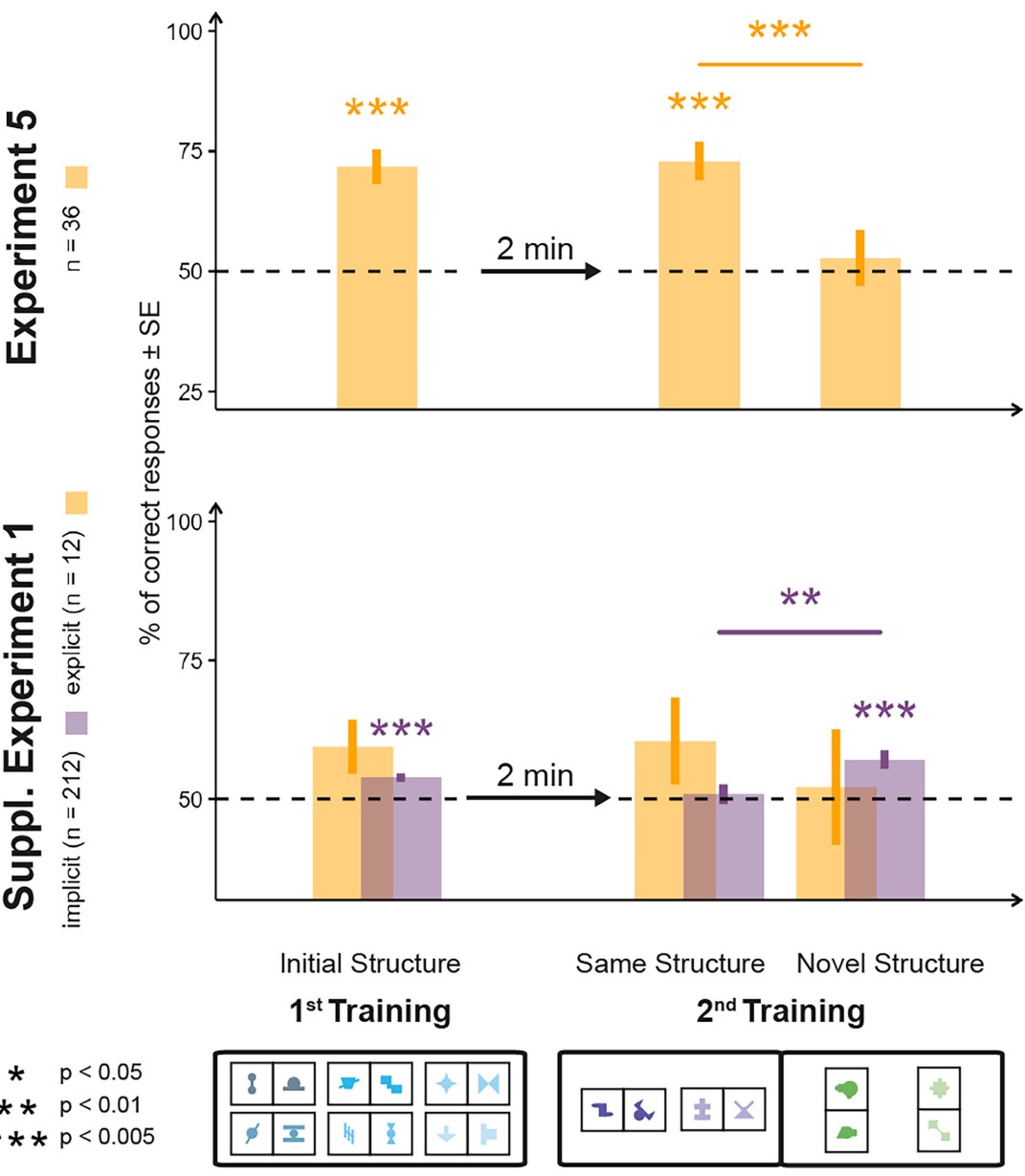

**Appendix 1—figure 1.** Results of familiarity tests in Experiment 5 and Supplementary Experiment 1. Test results of the two-alternative forced-choice (2AFC) trials in both experiments are grouped on the x-axis according to whether the trials used shapes of the first or the second training and, within the second training, whether the pair in the trial had the same or different orientation structure as inventory pairs in the first training. The y-axis represents the proportion of correct responses in the 2AFC test trials. Bars represent SEM, color coding indicates implicit and explicit subgroups of the participants. The horizontal dotted line denotes chance performance. Asterisks above bars denote significance levels from chance, while above lines, significance level comparing two conditions below the tips of the line.

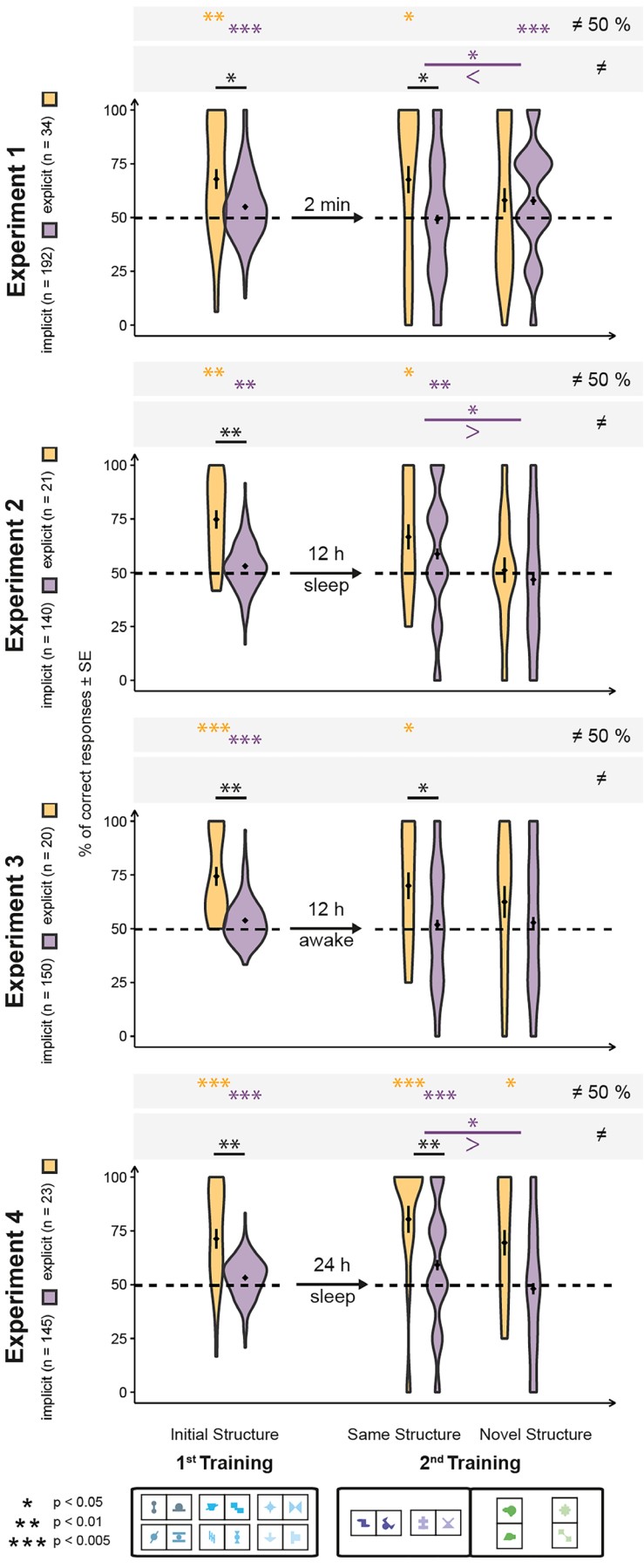

**Appendix 1—figure 2.** Violin plot version of *Figure 2*, depicting results of familiarity tests in Experiments 1–4. The width of the violins corresponds to a smoothed version of the relative frequency of values in the dataset. The markers within the violins denote the mean and standard error. Test results of the two-alternative forced-choice (2AFC) trials in all four experiments are grouped on the x-axis according to whether the trials used shapes of the first or the second training and, within the second training, whether the pair in the trial had the same or different orientation structure as inventory pairs in the first training. The y-axis represents the proportion of correct responses in the 2AFC test trials. Arrows and text between the test results related to the two trainings convey the condition and length of the break period. Legend of significance levels is shown in the lower-left corner. Signs of inequality below the comparison in the second training indicate the direction of effect.

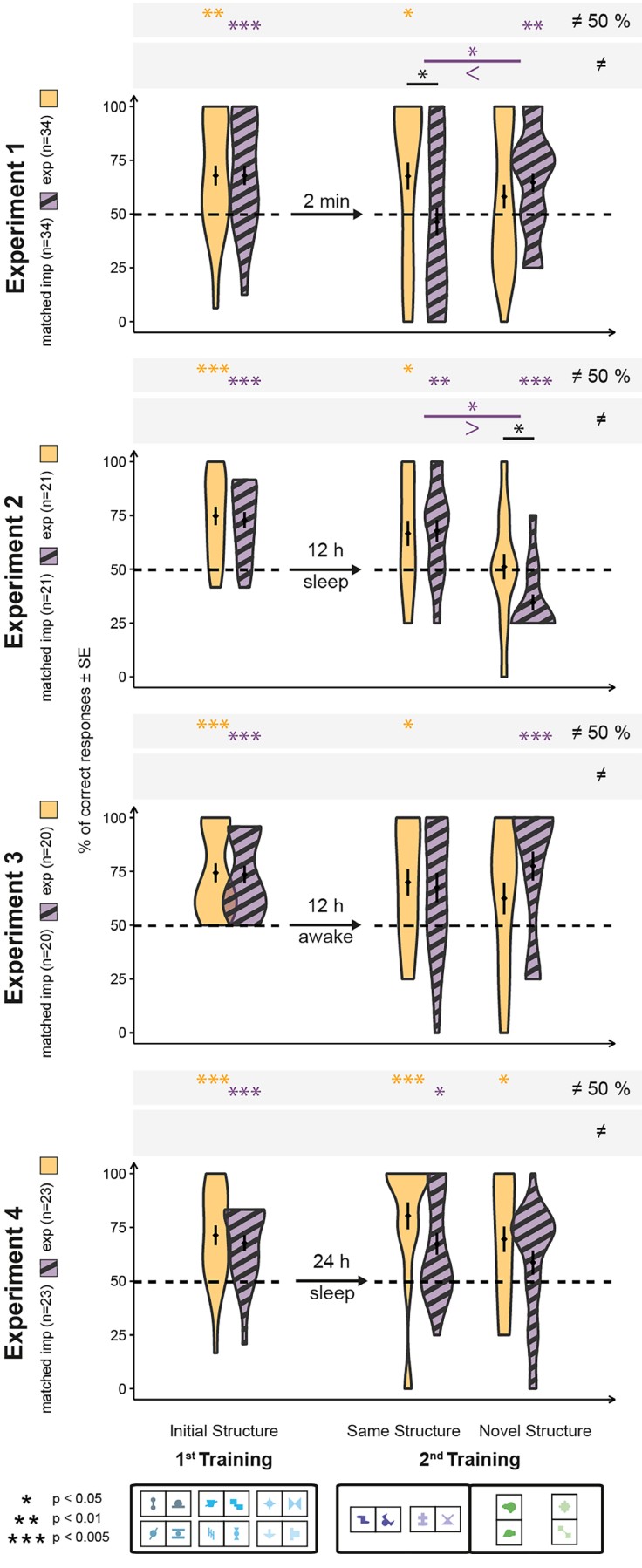

**Appendix 1—figure 3.** Violin plot version of *Figure 3*, depicting results of familiarity tests in the matched data for Experiments 1–4. The width of the violins corresponds to a smoothed version of the relative frequency of values in the dataset. The markers within the violins denote the mean and standard error. Test results of the two-alternative forced-choice (2AFC) trials in all four experiments are grouped on the x-axis according to whether the trials used shapes of the first or the second training and, within the second training, whether the pair in the trial had the same or different orientation structure as inventory pairs in the first training. The y-axis represents the proportion of correct responses in the 2AFC test trials. Arrows and text between the test results related to the two trainings convey the condition and length of the break period. Legend of significance levels is shown in the lower-left corner. Signs of inequality below the comparison in the second training indicate the direction of effect.

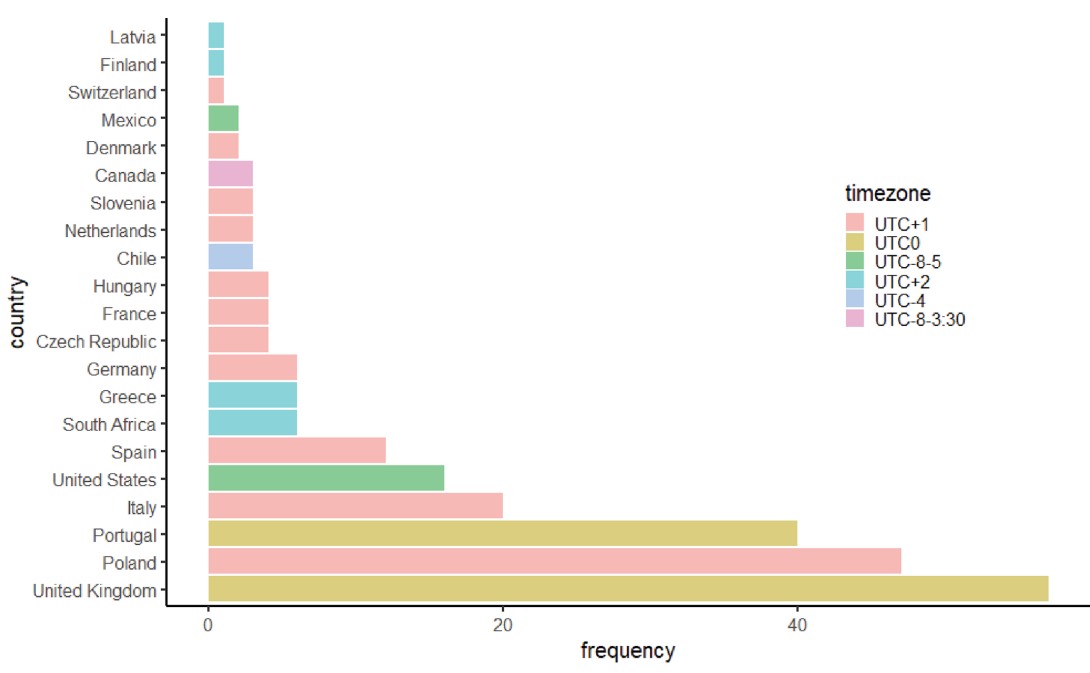

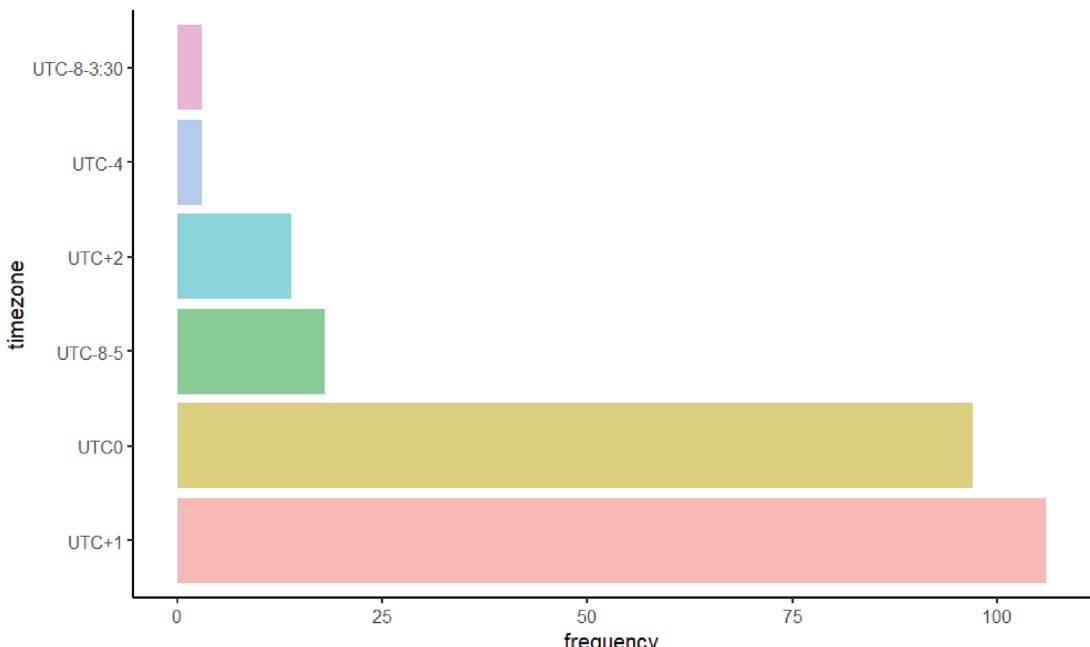

**Appendix 1—figure 4.** Frequency distribution of country of residence and time zone for participants in Experiment 1.

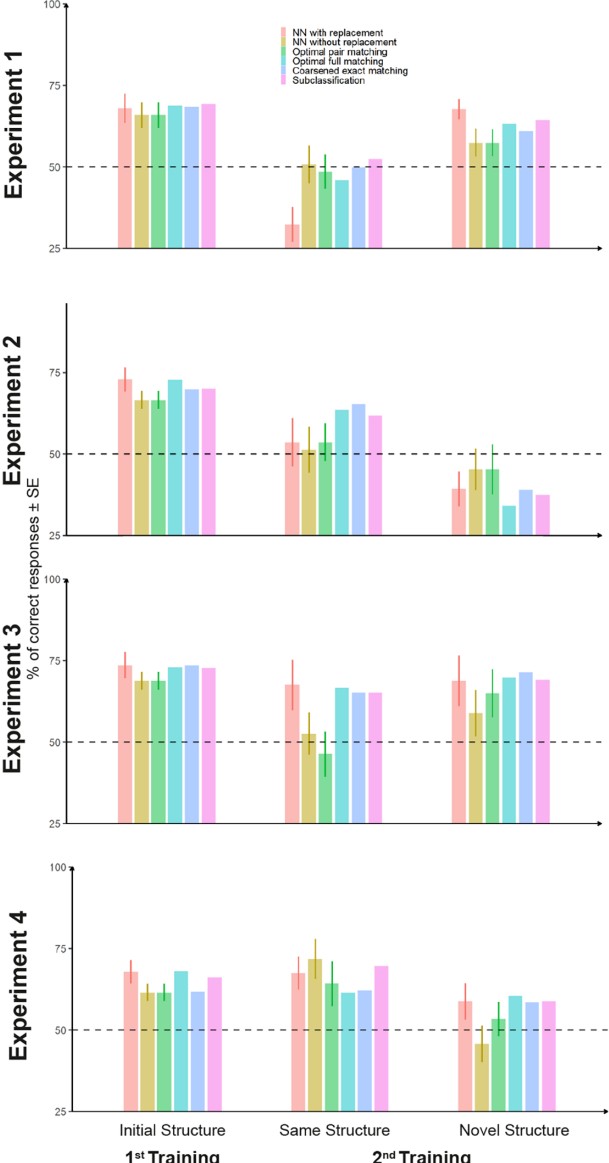

**Appendix 1—figure 5.** Results of familiarity tests in Experiments 1–4 for the matched implicit sample plotted for all tested matching algorithms. Test results of the two-alternative forced-choice (2AFC) trials in both experiments are grouped on the x-axis according to whether the trials used shapes of the first or the second training and, within the second training, whether the pair in the trial had the same or different orientation structure as inventory pairs in the first training. The y-axis represents the proportion of correct responses in the 2AFC test trials. The horizontal dotted line denotes chance performance. Color coding indicates the used matching algorithm.

