## [Editor Report · eLife Assessment]

This study investigates the conditions under which abstract knowledge transfers to new learning. It presents **convincing** evidence across a number of behavioral experiments that when explicit awareness of learned statistical structure is present, knowledge can transfer immediately, but that otherwise similar transfer requires sleep-dependent consolidation. The **valuable** results provide new constraints on theories of transfer learning and consolidation.

---

## [Referee Report · Reviewer #1 (Public review)]

Summary:

This paper investigates the effects of the explicit recognition of statistical structure and sleep consolidation on the transfer of learned structure to novel stimuli. The results show a striking dissociation in transfer ability between explicit and implicit learning of structure, finding that only explicit learners transfer structure immediately. Implicit learners, on the other hand, show an intriguing immediate structural interference effect (better learning of novel structure) followed by successful transfer only after a period of sleep.

Strengths:

This paper is very well written and motivated, and the data are presented clearly with a logical flow. There are several replications and control experiments and analyses that make the pattern of results very compelling. The results are novel and intriguing, providing important constraints on theories of consolidation. The discussion of relevant literature is thorough. In sum, this work makes an exciting and important contribution to the literature.

---

## [Referee Report · Reviewer #2 (Public review)]

Summary:

Sleep has not only been shown to support the strengthening of memory traces but also their transformation. A special form of such transformation is the abstraction of general rules from the presentation of individual exemplars. The current work used large online experiments with hundreds of participants to shed further light on this question. In the training phase participants saw composite items (scenes) that were made up of pairs of spatially coupled (i.e., they were next to each other) abstract shapes. In the initial training, they saw scenes made up of six horizontally structured pairs and in the second training phase, which took place after a retention phase (2 min awake, 12 hour incl. sleep, 12 h only wake, 24 h incl. sleep), they saw pairs that were horizontally or vertically coupled. After the second training phase, a two-alternatives-forced-choice (2-AFC) paradigm, where participants had to identify true pairs versus randomly assembled foils, was used to measure performance on all pairs. Finally, participants were asked five questions to identify, if they had insight into the pair structure and post-hoc groups were assigned based on this. Mainly the authors find that participants in the 2 minute retention experiment without explicit knowledge of the task structure were at chance level performance for the same structure in the second training phase, but had above chance performance for the vertical structure. The opposite was true for both sleep conditions. In the 12 h wake condition these participants showed no ability to discriminate the pairs from the second training phase at all.

Strengths:

All in all, the study was performed to a high standard and the sample size in the implicit condition was large enough to draw robust conclusions. The authors make several important statistical comparisons and also report an interesting resampling approach. There is also a lot of supplemental data regarding robustness.

Weaknesses:

My main concern regards the small sample size in the explicit group and the lack of experimental control.

---

## [Referee Report · Reviewer #3 (Public review)]

In this project, Garber and Fiser examined how the structure of incidentally learned regularities influences subsequent learning of regularities, that either have the same structure or a different one. Over a series of six online experiments, it was found that the structure (spatial arrangement) of the first set of regularities affected learning of the second set, indicating that it has indeed been abstracted away from the specific items that have been learned. The effect was found to depend on the explicitness of the original learning: Participants who noticed regularities in the stimuli were better at learning subsequent regularities of the same structure than of a different one. On the other hand, participants whose learning was only implicit had an opposite pattern: they were better in learning regularities of a novel structure than of the same one. However, when an overnight sleep separated the first and second learning phases, this opposite effect was reversed and came to match the pattern of the explicit group, suggesting that the abstraction and transfer in the implicit case were aided by memory consolidation.

---

## [Author Response]

The following is the authors’ response to the previous reviews.

**Public Reviews:**

**Reviewer #1 (Public review):**
Summary:This paper investigates the effects of the explicit recognition of statistical structure and sleep consolidation on the transfer of learned structure to novel stimuli. The results show a striking dissociation in transfer ability between explicit and implicit learning of structure, finding that only explicit learners transfer structure immediately. Implicit learners, on the other hand, show an intriguing immediate structural interference effect (better learning of novel structure) followed by successful transfer only after a period of sleep.Strengths:This paper is very well written and motivated, and the data are presented clearly with a logical flow. There are several replications and control experiments and analyses that make the pattern of results very compelling. The results are novel and intriguing, providing important constraints on theories of consolidation. The discussion of relevant literature is thorough. In sum, this work makes an exciting and important contribution to the literature.Weaknesses:There have been several recent papers which have identified issues with alternative forced choice (AFC) tests as a method of assessing statistical learning (e.g. Isbilen et al. 2020, Cognitive Science). A key argument is that while statistical learning is typically implicit, AFC involves explicit deliberation and therefore does not match the learning process well. The use of AFC in this study thus leaves open the question of whether the AFC measure benefits the explicit learners in particular, given the congruence between knowledge and testing format, and whether, more generally, the results would have been different had the method of assessing generalization been implicit. Prior work has shown that explicit and implicit measures of statistical learning do not always produce the same results (eg. Kiai & Melloni, 2021, bioRxiv; Liu et al. 2023, Cognition).The authors argued in their response to this point that this issue could have quantitative but not qualitative impacts on the results, but we see no reason that the impact could not be qualitative. In other words, it should be acknowledged that an implicit test could potentially result in the implicit group exhibiting immediate structure transfer.

We thank the reviewer for their feedback and added a statement in our discussion section acknowledging the possible effects of alternative measures of learning.

Given that the explicit/implicit classification was based on an exit survey, it is unclear when participants who are labeled "explicit" gained that explicit knowledge. This might have occurred during or after either of the sessions, which could impact the interpretation of the effects and deserves discussion.

We agree with the mentioned shortcoming in principle, although there are good methodological reasons for this, as discussed in our previous response. We added a statement on this topic to our discussion to make the potential issues and our reasoning in the design decision more transparent for the reader.

**Reviewer #2 (Public review):**
Summary:Sleep has not only been shown to support the strengthening of memory traces, but also their transformation. A special form of such transformation is the abstraction of general rules from the presentation of individual exemplars. The current work used large online experiments with hundreds of participants to shed further light on this question. In the training phase participants saw composite items (scenes) that were made up of pairs of spatially coupled (i.e., they were next to each other) abstract shapes. In the initial training, they saw scenes made up of six horizontally structured pairs and in the second training phase, which took place after a retention phase (2 min awake, 12 hour incl. sleep, 12 h only wake, 24 h incl. sleep), they saw pairs that were horizontally or vertically coupled. After the second training phase, a two-alternativesforced-choice (2-AFC) paradigm, where participants had to identify true pairs versus randomly assembled foils, was used to measure performance on all pairs. Finally, participants were asked five questions to identify, if they had insight into the pair structure and post-hoc groups were assigned based on this. Mainly the authors find that participants in the 2 minute retention experiment without explicit knowledge of the task structure were at chance level performance for the same structure in the second training phase, but had above chance performance for the vertical structure. The opposite was true for both sleep conditions. In the 12 h wake condition these participants showed no ability to discriminate the pairs from the second training phase at all.Strengths:All in all, the study was performed to a high standard and the sample size in the implicit condition was large enough to draw robust conclusions. The authors make several important statistical comparisons and also report an interesting resampling approach. There is also a lot of supplemental data regarding robustness.Weaknesses:My main concern regards the small sample size in the explicit group and the lack of experimental control.

We thank the reviewer for the valuable feedback throughout the review process. The issues mentioned here have been addressed in our previous response.

**Reviewer #3 (Public review):**
In this project, Garber and Fiser examined how the structure of incidentally learned regularities influences subsequent learning of regularities, that either have the same structure or a different one. Over a series of six online experiments, it was found that the structure (spatial arrangement) of the first set of regularities affected learning of the second set, indicating that it has indeed been abstracted away from the specific items that have been learned. The effect was found to depend on the explicitness of the original learning: Participants who noticed regularities in the stimuli were better at learning subsequent regularities of the same structure than of a different one. On the other hand, participants whose learning was only implicit had an opposite pattern: they were better in learning regularities of a novel structure than of the same one. However, when an overnight sleep separated the first and second learning phases, this opposite effect was reversed and came to match the pattern of the explicit group, suggesting that the abstraction and transfer in the implicit case were aided by memory consolidation.In their revision the authors addressed my major comments successfully and I commend them for that.
**Recommendations for the authors:**

**Reviewer #1 (Recommendations for the authors):**
We would encourage the authors to add text to the manuscript that acknowledges/discusses the two issues pointed out in our review.

We added relevant passages to the discussion section of the manuscript.

**Reviewer #2 (Recommendations for the authors):**
The authors have improved some sections of the manuscript and this is reflected in my assessment. The major weaknesses remain unchanged. Since my review is published alongside the paper, readers can make up their own mind regarding their severity.My only hard ask would be to add that the study was not preregistered into the main manuscript as I asked before! I am surprised that the authors are so reluctant to honestly state this fact....

We have not stated this fact in our manuscript until now since our understanding is that papers that report preregistered studies state and cite their preregistration in their method section, while any omission of such a statement by default conveys that no preregistration occurred. In fact, we cannot recall encountering papers with statements of no-preregistration in the literature. Nevertheless, we have no issue stating that our study was not preregistered and per the reviewer's request, we have added such an explicit statement in our manuscript.

**Reviewer #3 (Recommendations for the authors):**
* I strongly urge the authors to remove the Results sub-sections from Methods.

We thank the reviewer for highlighting this issue arising from our previous layout, which we decided to handle the following way. We re-labeledl the subsections in question as “Additional Analyses” to avoid confusion, we removed any redundant findings already reported in Results of the main text, and we moved a small number of more substantial findings from the Methods Section to the main text Results as requested. We believe that this solution constitutes the most readable option, as we do not clutter the main results with extensive sanity checks and results

of minor interest, while we also do not need to establish experiment-wise result sections in the Supplementary Materials, which would further disperse information interested readers might look for.

* Authors report that in Experiment 4 "Participants with explicit knowledge (n=23) show the same pattern of results as they did in Experiment 1", but that seems inaccurate, as they did learn novel pairs in Exp4 whereas they did not in Exp1. This can be seen in the figure and also in Methods-Results: "performing above chance for ... pairs of a novel structure (M=69.6, SE=5.9, d=0.69, t(22)=3.33 p=0.012, BF=13.6) in the second training phase"

We thank the reviewer for pointing out this error in our interpretation of the results and adjusted the section in question to better align with what our result actually shows.